

# Assessment of the effects of ischaemia/ hypoxia on angiogenesis in rat myofascial trigger points using colour Doppler flow imaging

Fangyan Jiang[1,*], Shuangcheng Yu[2,*], Haiqing Su[3] and Shangyong Zhu[1]

[1] Department of Medical Ultrasound, First Affiliated Hospital of Guangxi Medical University, Nanning, Guangxi, China
[2] Department of Radiology, Affiliated Minzu Hospital of Guangxi Medical University, Nanning, Guangxi, China
[3] Department of Medical Ultrasound, Affiliated Minzu Hospital of Guangxi Medical University, Nanning, Guangxi, China
[*] These authors contributed equally to this work.

Corresponding author
Shangyong Zhu,
zhushangyong2019@hotmail.com

## ABSTRACT

**Background & Aims**. Myofascial pain syndrome (MPS) is a common non-articular disorder of the musculoskeletal system that is characterized by the presence of myofascial trigger points (MTrPs). Despite the high prevalence of MPS, its pathogenesis, which induces the onset and maintenance of MTrPs, is still not completely understood. To date, no studies have investigated the changes in the biochemical milieu caused by ischaemia/hypoxia in the MTrP regions of muscle that are proposed in the integrated hypothesis. Therefore, this study investigated whether ischaemic/hypoxic conditions participate in the formation of active MTrPs and affect angiogenesis using colour Doppler flow imaging (CDFI).

**Methods**. Twenty-five Sprague-Dawley rats were randomly divided into a model group and a normal control group. A model of active MTrPs was established by a blunt strike combined with eccentric exercise. Enzyme-linked immunosorbent assays (ELISAs) were employed to detect the levels of HIF-1$\alpha$ and VEGF. Microvessel density (MVD) was evaluated using immunohistochemistry. CDFI was applied to observe the blood flow signals in the MTrPs, which were classified into four grades based on their strengths.

**Results**. Compared with the control group, the active MTrP group exhibited significantly higher HIF-1$\alpha$ and VEGF levels and MVD values. These differences were accompanied by increased blood flow signals. In the active MTrP group, the blood flow signal grade was positively correlated with the MVD ($P < 0.05$) and independently correlated with the VEGF level ($P < 0.05$) but was not correlated with the expression of HIF-1$\alpha$ ($P > 0.05$).

**Conclusion**. Ischaemic/hypoxic conditions may be involved in the formation of MTrPs. CDFI is useful for detection of the features of angiogenesis in or surrounding MTrPs via assessment of blood flow signals.

## INTRODUCTION

Myofascial pain syndrome (MPS) is a common non-articular disorder of the musculoskeletal system affecting as many as 15% of patients treated in general medical practice and up to 85% patients treated at pain management centres (*Fleckenstein et al., 2010*; *Srbely, Kumbhare & Grosman-Rimon, 2016*). MPS is characterized by the presence of myofascial trigger points (MTrPs), which are discrete, stiff and hyperirritable nodules in taut bands of skeletal muscle that can be palpated during a physical examination (*Shah et al., 2015*). MTrPs are clinically classified as active or latent. Many physicians currently make a definite diagnosis of MPS upon finding one or more MTrPs. Knowledge of the underlying aetiology of MTrPs is critical not only for preventing their development and recurrence but also for inactivating and eliminating existing MTrPs (*Bron & Dommerholt, 2012*). A consensus has been developed that muscle overuse, direct trauma or psychological stress lead to the development of MTrPs (*Simons, Travell & Simons, 1999*; *Shah et al., 2015*). Most individuals experience muscle pain as a result of trauma, injury, overuse, or strain at some point in their lives (*Shah et al., 2015*). If the muscle pain persists for a long period after resolution of the injury factors and if normal recovery is disrupted, MTrPs may develop.

Despite the high prevalence of MPS, its pathogenesis, which induces the onset and maintenance of MTrPs, is still not completely understood (*Jafri, 2014*). Currently, an intriguing possible mechanism mentioned by Simons, "The Integrated Trigger Point Hypothesis", is widely accepted by various researchers and postulates that an "energy crisis" perpetuates an initial, sustained, low-level muscle contraction, and a decrease in intramuscular perfusion is presumed to exist (*Simons, Travell & Simons, 1999*). These changes lead to local ischaemia, hypoxia, and insufficient ATP synthesis, which are responsible for increases in acidity and $Ca^{2+}$ accumulation and subsequent sarcomere contraction. When sarcomere contraction persists, local intramuscular perfusion may be decreased, and ischaemia/hypoxia can persist. This vicious cycle likely leads to the development of MTrPs (*Shah et al., 2015*; *Simons, Travell & Simons, 1999*).

According to previous studies, increased levels of pain and inflammatory biomarkers have been detected in the vicinity of active MTrPs (*Shah et al., 2008*; *Shah et al., 2005*; *Grosman-Rimon et al., 2016*; *Lv et al., 2018*). These findings objectively support Simons' integrated hypothesis. However, to date, no studies have investigated the changes in the biochemical milieu caused by ischaemia/hypoxia in the MTrP regions of muscle, and researchers have not yet clearly determined whether these changes are involved in the progression of MTrPs. Hypoxia increases the expression of HIF-1 α, a key oxygen concentration-dependent transcription factor, subsequently regulating the expression of its downstream target gene VEGF (*Pugh & Ratcliffe, 2003*). Discovery of this mechanism has confirmed that ischaemia/hypoxia regulates the process of angiogenesis (*Pugh & Ratcliffe, 2003*). In addition, VEGF has also been confirmed to be involved in muscle repair

mechanisms and capillary formation in skeletal muscle (*Olfert et al., 2010*; *Li et al., 2010*). Thus, the levels of HIF-1α and VEGF likely increase in response to active MTrPs.

In this study, an active MTrP model based on direct trauma was established in rats using a method described in a previous study (*Huang et al., 2013*). The aim of the present study was to compare the levels of HIF-1α, VEGF and microvessel density (MVD) between an active MTrP group and a normal control group to confirm the occurrence of ischaemia/hypoxia, as mentioned in the integrated hypothesis, and to investigate whether these indicators are associated with the grades of blood flow signals on ultrasound.

## MATERIALS AND METHODS

### Experimental animals

Twenty-five healthy male Sprague-Dawley rats that were 7 weeks old (body weight: 200–220 g) were purchased from the Animal Experiment Center of Guangxi Medical University (Nanning, China) and randomly divided into two groups: (1) a normal control group ($n = 10$) and (2) a model group ($n = 15$). All rats were housed in a pathogen-free animal facility with a controlled temperature of 22–24 °C and 42% humidity and maintained on a constant 12 h dark/12 h light cycle. The animals were provided free access to food and water. At the end of the experiment, all rats were euthanized by intraperitoneal injection of pentobarbital sodium (200 mg/kg), and tissue sections were obtained. The animal experiments were conducted in accordance with local protocols for the care and use of laboratory animals. The procedures were all approved by the Animal Experiment Center of Guangxi Medical University—also known as the Animal Ethics Committee of Guangxi Medical University—approved this study (Approval No: 201904013).

### Animal model

The active MTrP model was established as described by Huang et al through a blunt strike in combination with eccentric exercise for 8 weeks (*Huang et al., 2013*). Rats were anaesthetized by injection of 30 mg/kg pentobarbital sodium before being fixed on the board of a homemade striking device every Saturday. A blunt strike was administered to an area of the right proximal gastrocnemius that had been marked on the skin using a 1200 g stick freely dropped from a height of 20 cm with a kinetic energy of 2.352 J. On the second day (every Sunday), all injured rats underwent 90 minutes of eccentric exercise on a treadmill (SA101B, Jiangsu Saiangsi Biological Technology Co., Ltd., Nanjing, China) at a −16° downhill angle and speed of 16 m/min. Subsequently, the rats rested for 5 days a week without any intervention. All rats in the model group received this treatment for 8 weeks with 4 subsequent weeks of rest, while the rats in the control group did not undergo any intervention in this period.

### Ultrasound image processing

After modelling, the fur and skin covering the right gastrocnemius area were shaved and cleaned. Then, all rats underwent ultrasonographic examination using an Aplio 500 clinical ultrasound (US) system (Toshiba Medical System Corporation, Tochigi, Tokyo, Japan ) with a linear array transducer (5–14 MHz). The presence of MTrPs within

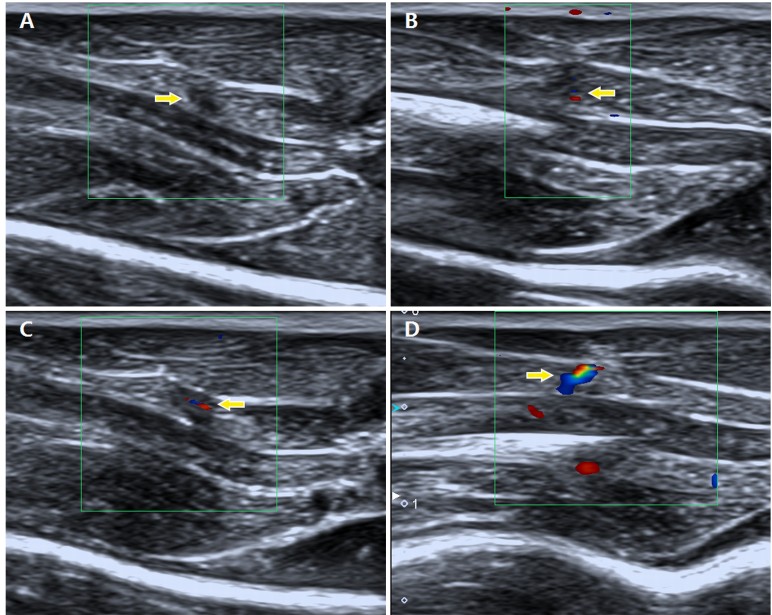

**Figure 1 Description of different grades of blood flow signals in the MTrPs observed using colour Doppler flow imaging.** (A) Grade 0, no blood flow signals; (B) grade I, one or two dot-like blood flow signals; (C) grade II, three dot-like or thin and short blood flow signals; and (D) grade III, one or more large and long blood flow signals.

the taut band was determined by two diagnostic sonographers with 10 and 16 years of experience using a combination of greyscale imaging and sonoelastography. According to previous studies (*Sikdar et al., 2009*; *Kumbhare, Elzibak & Noseworthy, 2016*; *Shankar & Reddy, 2012*), MTrPs are focal hypoechoic (darker) or hyperechoic (brighter) areas with heterogeneous echotextures on greyscale images and stiffer regions on sonoelastography than the surrounding tissues. In the present study, strain elastography (SE) was applied to identify the MTrPs using real-time colour elastography and greyscale US imaging. The hardness of tissues increased gradually, as shown by colours ranging from green (soft tissue) to blue (hard tissue). Colour Doppler flow imaging (CDFI) was applied to detect blood flow in the MTrPs. The blood flow signals were semiquantitatively classified into the following four grades based on the criteria reported by Adler (*Adler et al., 1990*): grade 0, no blood flow signal; grade I, one or two dot-like blood flow signals; grade II, three dot-like or thin and short blood flow signals; and grade III, one or more large and long blood flow signals (Fig. 1). When more than one MTrP was diagnosed within the gastrocnemius area using US imaging, the point with the richest blood flow signal in the referenced location was chosen.

## Electromyographic and histological assays

All rats were examined by electromyography (EMG) using an NTS-2000 instrument (Nuocheng Medical Co., Ltd., Shanghai). First, the right gastrocnemius area of each rat was completely exposed. The palpable taut band of muscle was marked. Then, an electrode was

inserted into the taut band to detect the local twitch responses and spontaneous electrical activity (SEA), which determined the presence of active MTrPs (*Huang et al., 2013*). After the SEA was determined, segments of muscle corresponding tissue were excised and fixed in formalin buffer for subsequent assessment of the histological changes in active MTrPs. Similarly, the rats in the control group underwent EMG, and histological assays were performed at the same positions.

## Microvessel density analysis

Muscle slices from the two groups of rats were stained with a cluster of differentiation 31 (CD 31) antibody to quantify the MVD (number of vessels). Paraffin sections were fixed and stained with haematoxylin-eosin (HE). The primary anti-CD31 antibody (1: 500, v/v) was added, and the sections were incubated overnight at 4 °C. After three 5-minute washes, the sections were incubated with a secondary anti-mouse IgG (1:2000) antibody at 37 °C for 1 h. Then, the samples were washed with PBS. The area with the highest microvascular numbers (hot spot) was selected under a × 40 optical field of view in three different fields as described by *Weidner et al. (1991)*. The average number of microvessels in the three selected fields at ×200 magnification was calculated.

## Enzyme-linked immunosorbent assay

Due to the extremely small sizes of MTrPs ($3-4 \text{ mm}^2$), homogenates were prepared from muscle tissue in the vicinity of MTrPs to increase the sensitivity of the test. The tissue samples were cut, weighed, frozen in liquid nitrogen and stored at $-80$ °C. Muscle tissue (100 mg) was rinsed with $1 \times$ PBS, homogenized in 1 ml of $1 \times$ PBS and stored overnight at $-20$ °C. After two freeze-thaw cycles were performed to disrupt the cell membranes, the homogenates were centrifuged at $5000 \times$ g for 5 minutes at $2-8$ °C. The supernatant was removed and assayed immediately. The levels of HIF-$1\alpha$ and VEGF were measured by enzyme-linked immunosorbent assay (ELISA) according to the manufacturer's instructions (Cusabio Biotech Co., Ltd., Wuhan, China).

## Statistical analysis

All data were analysed with SPSS 22.0 software (Chicago, IL, USA). The normality of the data was assessed using the Kolmogorov-Smirnov test. Descriptive statistics, including the HIF-$1\alpha$ and VEGF levels and the MVD values, are presented as the means and standard deviations. Independent-sample T tests were performed to verify the differences between the MTrP group and the normal control group. The correlations between the blood flow signal grades and the MVD values, VEGF levels and HIF-$1\alpha$ levels were determined by calculating the Spearman rank correlation coefficients. For all tests used, statistical significance was set at a value of $P < 0.05$.

# RESULTS

Of the 15 model rats, one died unexpectedly due to complications of anaesthesia. Therefore, 24 rats survived in the present study (model group: $n = 14$, normal group: $n = 10$).

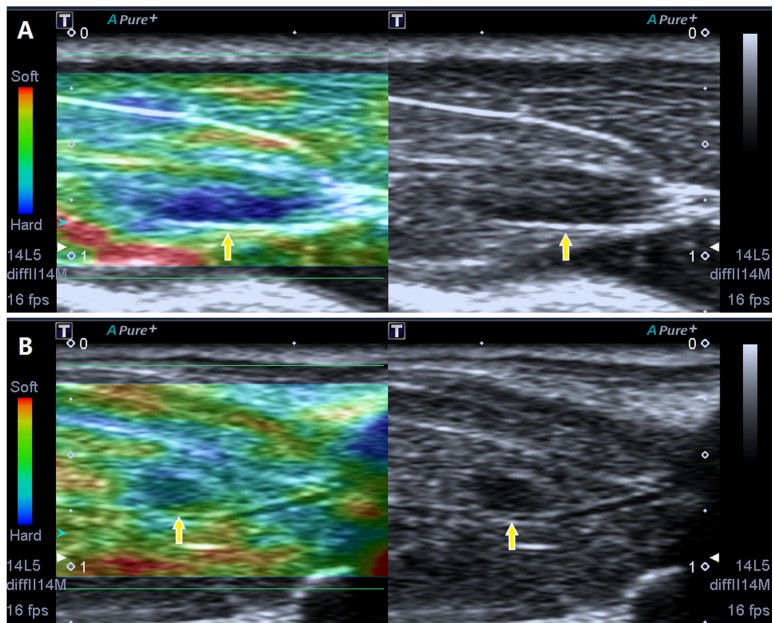

**Figure 2  Screenshot of the sonoelastography modes.** Strain elastography detection of a stiff focal region: the images presented in (A) and (B) are entirely covered in blue or are mostly blue with little green compared to the adjacent normal muscle tissue.

## Ultrasound findings

Via B-mode imaging, MTrPs were visualized in all rats of the model group within the taut band of the right gastrocnemius muscle in an area consistent with the location of the palpable nodule identified during physical examination. No MTrPs were observed in the control group. The great majority of MTrPs showed an ellipsoidal focal hypoechoic region, and 2 rats exhibited a hyperechoic region. Among the animals, 4 rats presented with more than one hypoechoic region, and the point of the greatest blood flow signal was chosen. The average MTrP size was $3.3 \pm 0.18$ mm$^2$. In addition, SE imaging consistently detected a stiff focal region that was entirely covered in blue or was mostly blue with little green, which are consistent with the results described in a previous study (*Kumbhare, Elzibak & Noseworthy, 2016*) (Fig. 2).

Based on the CDFI examination, the vessels in and around MTrPs exhibited higher blood flow than corresponding normal muscle sites, the latter of which showed no blood flow signals. A detailed description of the data obtained from the rats in the MTrP group is provided in Table 1.

## Electromyography results and pathological features

The taut bands of the muscles exhibiting abnormal changes on ultrasound were verified to show spontaneous electrical activity and local twitch responses using an electromyographic device. The rats in the normal control group showed no electromyographic activity (Fig. 3).

Examination of the pathological sections of MTrP areas revealed variations in the shapes and sizes of the muscle fibres. Specifically, muscle fibres became thinner at both ends and

**Table 1  Results of blood flow signal grading.**

| Grade | Criterion | No. ($n = 14$) |
|---|---|---|
| 0 | No blood flow signals | 2 |
| I | One or two dot-like blood flow signals | 5 |
| II | Three dot-like or a thin- and short-like blood flow signals | 4 |
| III | One or more large and longer blood flow signals | 3 |

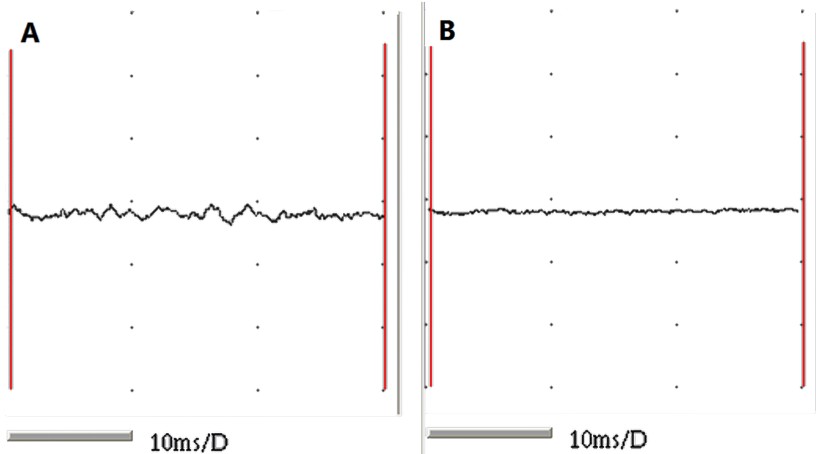

**Figure 3  EMG recordings from the two groups.** (A) Muscle fibres from MTrPs presenting with spontaneous electrical activities and (B) muscle fibres from normal controls showing no EMG activity.

swelled in the middle. Contraction nodules were present in the longitudinal section along with local widening of the muscle fibre gaps. However, the sizes and shapes of muscle fibres from normal controls were uniform (Fig. 4). These findings are consistent with results reported in the published literature (Zhang et al., 2017).

## Microvessel density detection

As shown in Fig. 5, the cytoplasm of vascular endothelial cells in the gastrocnemius muscles of rats was stained brownish-yellow with an antibody against the CD31 protein. The MVDs of the model and control groups observed under ×200 optical magnification were 123.64 ± 9.56 and 84.70 ± 13.46, respectively. The MVD of the MTrP group was significantly higher than that of the control group ($P = 0.000$). Additionally, comparison of the vascular morphology of the two groups revealed that the blood vessels in the MTrP group were thicker and less regularly distributed than the vessels in the control group.

## HIF-1α and VEGF expression in the two groups

HIF-1α and VEGF levels were measured in muscle tissue homogenates from the two groups of rats. As shown in Fig. 6, significantly higher levels of HIF-1α and VEGF were detected in the MTrP group than in the normal control group ($P < 0.05$).

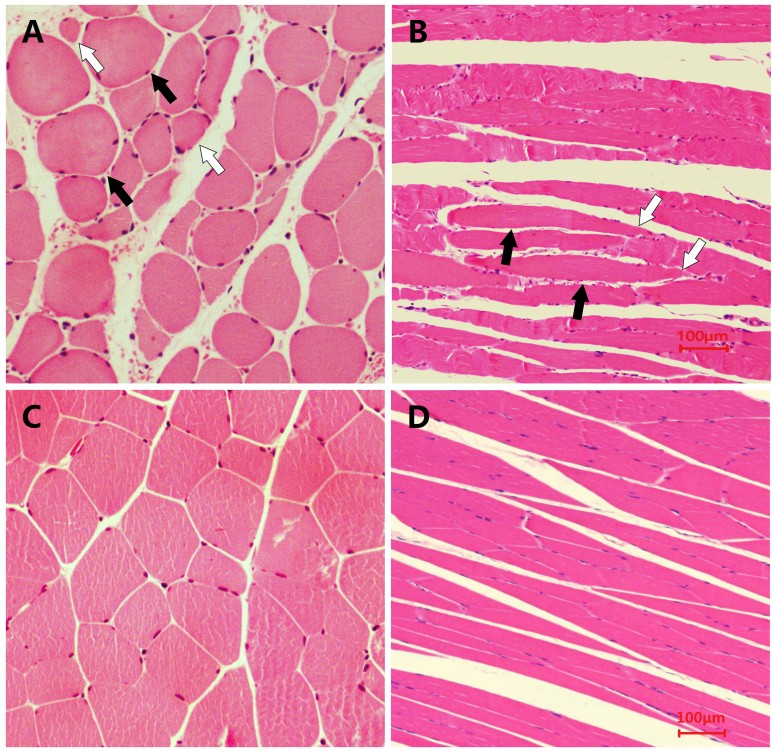

**Figure 4 Microscopic views of HE staining of muscle fibres from the two groups of rats (400×).** The histological changes in MTrPs are depicted in (A) (cross-section) and (B) (longitudinal section), and show that muscle fibres became thinner at both ends and swelled in the middle to show the contraction nodules in longitudinal sections, along with local widening of the muscle fibre gap. The histologically normal muscle fibres are similarly shown in (C) and (D), which reveal a uniform size and shape of muscle fibres.

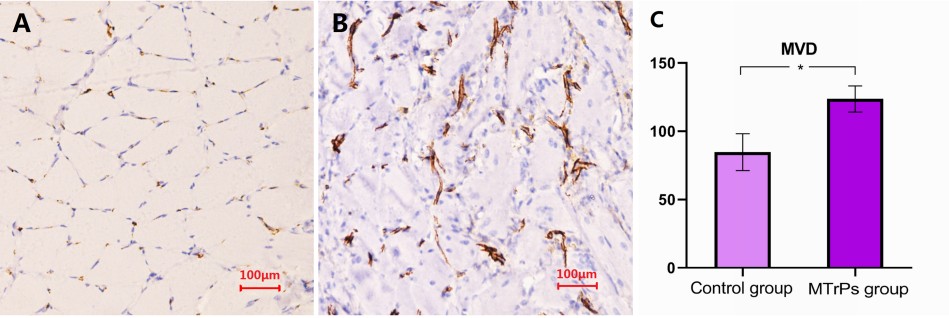

**Figure 5 Comparison of the MVD in rat muscle fibres between the MTrP rats and normal controls.** (A and B) Images of IHC staining for CD31 in rats from the control group and the MTrP group, respectively. (C) Comparison of the MVD in muscle fibres from rats in the two groups; *$P < 0.001$.

## Analysis of correlations between blood flow signals and HIF-1α levels, VEGF levels and microvessel densities in the myofascial trigger point group HIF-1α and VEGF expression in the two groups

In the active MTrP group, linear correlation analysis were performed between VEGF levels and MVDs and the blood flow signal grades. The correlation coefficients were 0.595 and

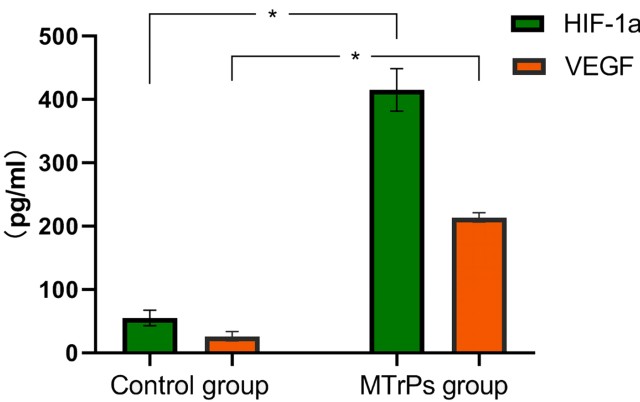

**Figure 6  Expression of HIF-1α and VEGF in two groups of rats.** An asterisk (*) indicates a significant difference between the MTrPs group and normal control group ($P < .001$).

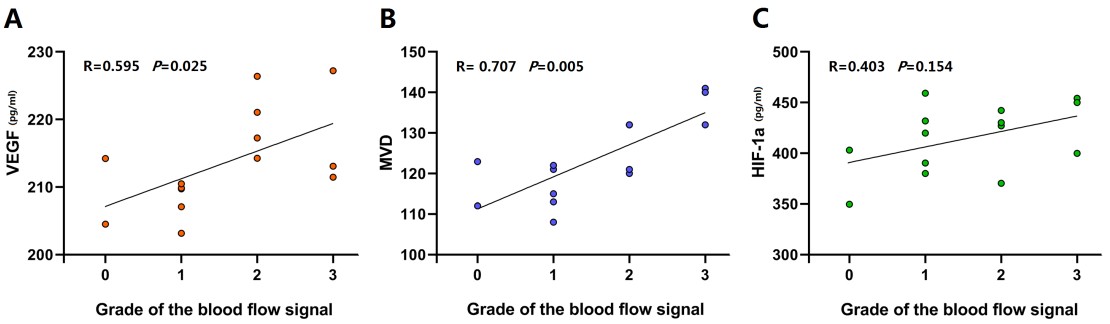

**Figure 7  Linear correlation plots between blood flow signals and HIF-1α levels, VEGF levels and MVD performed in the active MTrP group.** (A and B) Linear correlation analysis showed between VEGF levels and blood flow signal grades, MVD and blood flow signal grades a coefficient of correlation ($r$) > 0.5 ($p <$ 0.05). (C) A significant correlation between HIF-1 α levels and blood flow signal grades was not observed ($p > 0.05$).

0.707, respectively. A significant correlation between HIF-1α levels and blood flow signal grades was not observed (Fig. 7).

## DISCUSSION

The present study was conducted to determine whether ischaemia/hypoxia is involved in the formation of MTrPs, as proposed in the integrated hypothesis, and to investigate the underlying effect on angiogenesis. In the present study, an active MTrP rat model was successfully established by blunt-force trauma in combination with eccentric exercise, as confirmed through palpation of taut bands of muscle, SEA and a local twitch response according to the modified criteria defined by *Simons, Travell & Simons (1999)* and consistent with the histopathological and ultrasonographic changes reported in previous studies (*Zhang et al., 2017*; *Sikdar et al., 2009*; *Kumbhare, Elzibak & Noseworthy, 2016*; *Shankar & Reddy, 2012*). The preliminary findings of this study were as follows: (1) significantly higher HIF-1α and VEGF levels and higher MVDs were observed in rats with

active MTrPs than in normal controls, and (2) blood flow signal grade detected using CDFI was positively correlated with VEGF expression and MVD. These results are consistent with the intramuscular changes associated with an ischaemia-induced inflammatory microenvironment, providing objective evidence supporting the integrated hypothesis of MTrPs.

Simons' integrated hypothesis postulates that initial sustained low-level sarcomere contraction shapes local hypercontractions and compresses capillaries to cause local ischaemia/hypoxia, which is accompanied by increased local metabolic demands and leads to an energy crisis. This crisis contributes to the formation of MTrPs (*Simons, Travell & Simons, 1999*). In the presence of local ischaemia/hypoxia, tissue damage, or both, the biochemical milieu surrounding MTrPs becomes increasingly complex. Elevated levels of local biomarkers associated with pain and inflammation, including inflammatory mediators, neuropeptides, catecholamines, and cytokines, have been detected in the vicinity of active MTrPs compared with latent MTrPs or non-MTrP areas (*Shah et al., 2005*; *Shah et al., 2008*). Muscle pain may be associated with ischaemia-induced inflammation in taut bands (*Gerwin, 2001*). Furthermore, a recent study reported higher serum levels of inflammatory and anti-inflammatory biomarkers, as well as the growth factors fibroblast growth factor-2 (FGF-2), platelet-derived growth factor (PDGF), and VEGF, in an MPS group compared with a non-MPS control group (*Grosman-Rimon et al., 2016*). Thus, MTrPs may represent an ischaemia-induced inflammatory microenvironment.

In the present study, higher levels of HIF-1α and VEGF were detected in the active MTrP group than in the normal control group. This finding indicates that ischaemia/hypoxia is involved in the formation of active MTrPs. HIF-1α is a major regulator of basic adaptive responses to hypoxia (*Pugh & Ratcliffe, 2003*). HIF-α is generally present at low levels in normoxic rodent tissues and may not be detectable even in areas of physiological hypoxia (*Pugh & Ratcliffe, 2003*; *Rosenberger et al., 2002*). When systemic hypoxia occurs or when tissue ischaemia increases, HIF-α expression and transcriptional activity increase. Hypoxia-inducible factors orchestrate the adaptive responses of endothelial cells to changes in oxygen tension by controlling survival, metabolism, and angiogenesis (*Potente, Gerhardt & Carmeliet, 2011*). Metabolic regulators control vessel growth by stimulating angiogenesis under hypoxic conditions to prepare tissues for oxidative metabolism (*Potente, Gerhardt & Carmeliet, 2011*). HIF-α subsequently regulates the expression of its downstream target gene VEGF (*Pugh & Ratcliffe, 2003*). VEGF, a major mediator of pathological angiogenesis with a predominant role in both tumour-induced and inflammation-induced angiogenesis (*Ferrara et al., 1996*; *Carmeliet & Jain, 2000*), directly regulates blood vessel formation (*Potente, Gerhardt & Carmeliet, 2011*). VEGF also participates in skeletal muscle injury and repair processes by increasing capillary formation in skeletal muscle and restoring blood flow to injured tissue, increasing the efficiency of skeletal muscle repair (*Olfert et al., 2010*; *Best, Gharaibeh & Huard, 2013*). In addition, the immunohistochemical staining conducted in the present study revealed an obvious increase in the MVD in the active MTrP group compared with the normal control group, directly confirming that angiogenesis was induced in zones of active MTrPs.

Currently, there is not a consensus in the literature regarding tissue perfusion in MTrPs. Although ischaemia/hypoxia may be an important factor contributing to trigger point formation, the associated microvascular changes have yet to be thoroughly elucidated. Previous studies seeking to describe blood vessels in MTrPs have focused mainly on the initiation of MTrPs induced by hypercontraction-mediated compression of capillary/venous beds. Studies on the microvascular changes to enable maintenance of MTrPs have not been performed. MPS is a chronic inflammatory disorder, and MTrPs are clinically classified as active or latent. Patients are considered to have active MTrPs if they have consistently experienced pain within the last 3 months. During this period, prolonged ischaemia/hypoxia causes vessels to adjust their shapes and functions to meet the changing oxygen demands of the tissue (*Potente, Gerhardt & Carmeliet, 2011*). In addition, it might stimulate the release of bradykinins, cytokines, substance P and inflammatory biomarkers in the muscle. The processes of chronic inflammation and angiogenesis are intimately linked and occur together (*Costa, Incio & Soares, 2007*). Hypoxia is a common stimulus of both processes and results in increased production of growth factors (*Carmeliet, 2005*; *Yancopoulos et al., 2000*). Upregulation of the transcription factor in response to blood flow ensures remodelling of the vasculature to adapt the vascular pattern to local tissue needs (*Potente, Gerhardt & Carmeliet, 2011*). Therefore, a logical hypothesis is that angiogenesis is increased in areas of active MTrPs compared with normal control areas.

Interestingly, this preliminary study confirmed an increase in the number of blood vessels and the presence of enlarged blood vessels in or around active MTrPs compared with normal muscle sites, consistent with previous reports (*Sikdar et al., 2010*; *Ballyns et al., 2011*; *Grosman-Rimon et al., 2016*). These findings may provide additional insights that will help researchers elucidate the pathophysiological mechanisms underlying the development of MTrPs. By analysing the changes in microvessel numbers, vascular morphology, and blood flow signals in the zones of MTrPs, we identified two intriguing phenomena: an increased vascular volume and an increased number of angiogenic vessels. Increased vascular volume has been observed using lumped-parameter compartment models of the skeletal circulation (*Sikdar et al., 2010*). In the current study, our observations of more microvessels, thicker blood vessels, and richer blood flow signals in MTrPs than in normal tissues confirmed the increased vascular volume in MTrP tissues compared with non-MTrP control tissues. Furthermore, the MVDs determined in the present study intuitively revealed increased numbers of angiogenic vessels. However, when analysing pathological angiogenesis, researchers should focus on the functional qualities of vessels and their effects on local metabolism rather than on vessel quantity alone (*Potente, Gerhardt & Carmeliet, 2011*). Such information would be helpful in explaining the differences between MVD and local ischaemia in the areas of trigger points, which deserves further exploration.

Although the diagnosis of MTrPs was determined using conventional US combined with an SE examination, the aim of this study was to investigate the features of blood flow signals. To date, a few groups (*Sikdar et al., 2009*; *Sikdar et al., 2010*; *Ballyns et al., 2011*) have used Doppler imaging to examine the vasculature of regions surrounding MTrPs, and highly resistive vascular beds have been observed in and around the MTrPs. However, no studies have assessed the characteristics of angiogenesis at trigger points. CDFI is the

most common tool used to sensitively visualize blood flow. In the present study, stronger blood flow signals were observed in the MTrP group than in the control group. In addition, conspicuous blood vessels were observed at 12 of the 14 sites in the MTrPs group, while no blood flow signals were detected in the control group. Intriguingly, the blood flow signals in the zones of MTrPs were mainly of grades I–II (9/14), with 3 cases of grade III, suggesting that the majority of subjects presented dot-like or short blood flow signals. Only two animals exhibited large and long blood vessels that passed through the trigger points, consistent with previous reports (*Sikdar et al., 2010*).

In addition, the blood flow signal grade was positively correlated with the VEGF level and MVD in the active MTrP group. The correlation with the VEGF level was weaker than the correlation with the MVD. Three explanations for this weak correlation are proposed. First, the MVD not only is a standard indicator of the number of angiogenesis events but also intuitively reflects the morphology and distribution of microvessels (*Magnon et al., 2007*). The neovascularization of MTrPs was greater than that of normal muscle tissue in the present study. In addition, blood vessel formation and expansion and irregular blood vessel distribution were observed in the active MTrPs. The pattern obviously differed from the uniform and regular distribution of blood vessels in the control tissues. Second, VEGF is only one factor that promotes angiogenesis; other growth factors, such as FGF-2 and PDGF, are also involved (*Grosman-Rimon et al., 2016*). VEGF expression generally correlates closely with MVD (*Zou et al., 2016*; *Maria et al., 2005*). Third, time may elapse between VEGF expression in MTrPs and subsequent blood vessel formation that is detectable using CDFI.

There were some limitations of the present study. First, we adopted a tissue homogenization method to measure the levels of HIF-$\alpha$ and VEGF with ELISA to increase the sensitivity of the test, which might affect the detection of the levels of the target proteins. The serum levels were not investigated to compare with the current results. Second, both hypoechoic and hyperechoic MTrP regions were visualized using ultrasonography in the current study. The differences in the echogenicity of the MTrP regions on ultrasonography suggest differences in local tissue density features. The current study did not determine an association between a unique echo pattern and a specific MTrP histopathology. Identification of MTrPs using US imaging must be substantially improved. Third, due to different purposes in the studies, the time-course analysis was not scheduled in the present study. Time-course change in this model would be helpful for better understanding of the pathophysiological mechanisms of MTrPs, which requires further research.

## CONCLUSIONS

As shown in the present study, significantly higher HIF-1$\alpha$ and VEGF levels and MVDs were observed in the active MTrP group than in the normal control group, indicating that these factors may be affected by muscle ischaemia/hypoxia. These findings support the hypothesis that ischaemia/hypoxia is involved in the formation of MTrPs. In addition, the blood flow signal grade was positively correlated with the VEGF level and MVD in the

active MTrP group, suggesting that CDFI can be used to detect the features of angiogenesis in or surrounding MTrPs by assessing the blood flow signals.

### Funding
This work was supported by the Self-Funded Research Project of Guangxi Zhuang Autonomous Region National Health Planning Commission (No. Z20200569). The funders had no role in study design, data collection and analysis, decision to publish, or preparation of the manuscript.

### Grant Disclosures
The following grant information was disclosed by the authors:
Self-Funded Research Project of Guangxi Zhuang Autonomous Region National Health Planning Commission: Z20200569.

### Competing Interests
The authors declare that they have no competing interests.

### Author Contributions
- Fangyan Jiang and Shuangcheng Yu performed the experiments, analyzed the data, prepared figures and/or tables, authored or reviewed drafts of the paper, and approved the final draft.
- Haiqing Su analyzed the data, prepared figures and/or tables, and approved the final draft.
- Shangyong Zhu conceived and designed the experiments, analyzed the data, prepared figures and/or tables, authored or reviewed drafts of the paper, and approved the final draft.

### Animal Ethics
The following information was supplied relating to ethical approvals (i.e., approving body and any reference numbers):

The Animal Experiment Center of Guangxi Medical University—also known as the Animal Ethics Committee of Guangxi Medical University—approved this study (Approval No: 201904013).

### Data Availability
The data are available as Supplementary Files

### Supplemental Information
Supplemental information for this article can be found online at http://dx.doi.org/10.7717/peerj.10481#supplemental-information.

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
