# Peer review of "Assessment of the effects of ischaemia/ hypoxia on angiogenesis in rat myofascial trigger points using colour Doppler flow imaging"

_PeerJ, doi:10.7717/peerj.10481_

## Round 0.1 · original submission · Major Revisions

As you will learn from the reviews, the referees identified several conceptual and methodological problems. The paper needs to be revised and significantly improved according to the reviewers' recommendations for future editorial considerations. For resubmission, a thorough point-by-point response to the reviews is needed. Also, note that proofreading and professional language editing in the manuscript should be carried out carefully.

Reviewer 1 ·

Basic reporting

The paper is written in a clear English, even if some corrections are needed.
Literatrure references are provided to illustrate the background, even if the authors fail to adeguately quote them in the text.
The structure of the article is well designed except the organization/numbering of figures which does not correspond to the text.
In detail:
1) In the keywords please correct: Ischemia – Hypoxia.
2) Figures: there is no correspondence between the numbering in the text (Figures 1-6) and the single figures (numbered 1-16). Please provide numbering and lettering of panels and appropriate titles and legends of the 6 figures.
3) References at lines 296-297 have no correspondence in the reference list. Please check names/surnames for accuracy.

Experimental design

The paper by Jiang and co-workers reports that trauma and overuse induced ischemia-hypoxia conditions are involved in the formation of myofascial trigger points (MTrPs), then responsable of muscular pain. Color doppler flow imaging (CDFI) is associated with neovascularization markers as HIF-1a and VEGF. The paper demonstrates that CDFI by assessing the blood flow signal through a not bloody external procedure can detect the features of angiogenesis in or surrounding MTrPs.
The paper is interesting and the results reported sound enough.
Major comments:
1) Could the authors give a rationale for the time used in the model? “All rats in the model group were treated in this way for 8 weeks with a subsequent 4 weeks of rest, while the rats in the control group did not undergo any intervention in this period.” Did the author performed a time course study to evaluate neovascular responses following injury? The time for angiogenesis occurrence is usually shorter (few days-1 week) and probably the authors uasing shorter times can reach more striking results respect to the ones reported here (i.e. figure 6 and table 2).
2) Soon after the muscle trauma (1200 g stick freely dropped from a height of 20 cm!) it is supposed to have tissue damage due to compression, a strong inflammatory reaction and platelet activation. The authors should consider the role played by angiogenic factors released by inflamamtory cells and platelets, again at short times after the injury.
3) HIF-1a and VEGF measurements by ELISA: please check since in the MM section it is reported that they are done on tissue homogenates, while in results it is reported serum levels.

Validity of the findings

The findings have enough impact and novelty.
All the data reported statistically sound.
Conclusions are well stated on the base of the reported data. Authors try to speculate and give possible explanations about the less robust data.

Additional comments

Please provide rationale and explanation to the above requests and improve the editing/organization of the mansucript.

Reviewer 2 ·

Basic reporting

I think the manuscript might meet the standard after major revision.

Experimental design

Methods: Microvessel density- it is written ‘MVD of the gastrocnemius muscles of rats from the two groups was detected by hematoxylin150 eosin (HE) staining’. MDV was analyzed by CD31 immunostaining. Such mistakes are not good for paper. What is the hotspot method, and how it was used in a tissue section to determine MVD?

Validity of the findings

No comment

Additional comments

Title: The effect of ischemia – hypoxia in rats on angiogenesis in myofascial trigger points assessed by color Doppler flow imaging
In this study, Jiang et al. investigated about the myofascial pain syndrome, which is characterized by the presence of myofascial trigger points (MTrPs). Specifically, their aim was to check the hypothesis proposed by Simons regarding the development of MTrPs in MPS. In the hypothesis proposed by Simons, local hypoxia plays an important role in the development of MTrP. In this paper, the authors showed increased angiogenesis at the areas of MTrP. Since increased angiogenesis suggests better tissue perfusion and MTrPs are developed due to hypoperfusion, I think the findings shown in the paper contradict Simons hypothesis. In the areas of MTrPs, vessels show some changes. The authors might investigate such vascular changes and correlate with angiogenesis event to explain the development of MTrPs or should fully explain what’s mechanism is involved there compared with the previous knowledge.
There are also some serious flaws in the paper:
1. HIF-1α ELISA: in the result section, the authors stated that ELISA was done with serum. But in the Methods section, they stated that muscle tissue was used for ELISA. Such discrepancy is not good for a paper. Also, the authors need to mention the tissue lysis method, because that might affect the quantity of target protein.
2. Description of results: Description of results is very incomplete. It is very difficult to correlate what is written in the result and the figures.
3. Figures: Figures are usually arranged in a paper according to the story. One figure tells 1 section of the story. In a figure, several sections are arranged as (A), (B) or (C) etc. Here 17 figures are shown in a random way, which is very difficult to follow. Some figures do not corroborate with the results. For example, in the results of HIF-1α ELISA, figure 6 is referred. But figure 6 is ‘the screenshot of the sonoelastography modes’.
4. Figure legends need to be amended. First, arrange the figures more carefully according to the story, and then write the figure legends. Figure legends should explain how we should understand the figures.
5. Discussion should be re-written completely.

---

## Round 0.2 · Major Revisions

There are valid concerns from reviewer 2 about methodological issues, especially the time-course analysis and language editing. Therefore, the paper did not achieve the priority necessary for publication. Please revise the paper in accordance with the reviewers' 2 suggestions.

Reviewer 1 ·

Basic reporting

The manuscript is cleary written and hypothesis and experimental data reported and quoted according to conventional scientific procedures.

Experimental design

The experimental desing is clearly reported according to ethycal and technical standads. Methods are described in a sufficient manner.

Validity of the findings

All the data are robust, controlled and statistically sound. Conclusions are well stated

Additional comments

All the referees' requests and comments have been satisfied and the mansucript corrected for English style and grammar.
The revised version is thus improved.

Reviewer 2 ·

Basic reporting

There are sometimes careless mistakes.

Experimental design

Time-course analysis should be necessary.

Validity of the findings

It is difficult to evaluate whether the results are reasonable.

Additional comments

Title: The effect of ischemia – hypoxia in rats on angiogenesis in myofascial trigger points assessed by color Doppler flow imaging
In this paper, the authors showed increased angiogenesis in the areas of MTrP. However, they speculated that the initiation of the MTrP is hypoperfusion, which was not verified in the present study. Since time-course analysis was not scheduled, it is very difficult whether their insisted theory is reasonable or not. They can investigate every parameter, especially blood flow with doppler flow imaging, from before and after making the disease model.
The relationship between blood flow signals and other parameters should be shown in the scatter graph because the correlation might be very weak.
It is unbelievable to see the title name is wrongly written.

---

## Round 0.3 · Minor Revisions

The reviewer is favorable towards the revision and the responses to comments. However, an additional issue identified by the reviewer is that of incorporating a discussion on the time-course change in the model. The manuscript is recommended to revise with reviewer comments for further editorial consideration.

Reviewer 2 ·

Basic reporting

Grammar and word mistakes are all improved.

Experimental design

The authors should include about time-course change in this model and the necessity of it will be discussed in Discussion. It may be added to the limitation.

Validity of the findings

Further sound investigation strengthens this study's evidence.

Additional comments

No more comments.

---

## Round 0.4 · accepted · Accept

The current revision and author’s response to previous comments are acceptable.